# Text Before Vision: Staged Knowledge Injection Matters for Agentic RLVR in Ultra-High-Resolution Remote Sensing Understanding

**Fengxiang Wang** [1 2]  **Mingshuo Chen** [1]  **Yueying Li** [1]  **Yulin Wang** [3 *]  **Yajie Yang** [4]  **Yuhao Zhou** [5]
**Di Wang** [6 7]  **Yifan Zhang** [8]  **Haoyu Wang** [3]  **Haiyan Zhao** [3]  **Hongda Sun** [9]  **Jun Song** [8]
**Long Lan** [1]  **Jing Zhang** [6 *]  **Wenlong Zhang** [2 *]  **Bo Du** [6]

## Abstract

Multimodal reasoning for ultra-high-resolution (UHR) remote sensing (RS) is usually bottle-necked by visual evidence acquisition: the model necessitates localizing tiny task-relevant regions in massive pixel spaces. While Agentic Reinforcement Learning with Verifiable Rewards (RLVR) using zoom-in tools offers a path forward, we find that standard reinforcement learning struggles to navigate these vast visual spaces without structured domain priors. In this paper, we investigate the interplay between post-training paradigms: comparing Cold-start Supervised Fine-Tuning (SFT), RLVR, and Agentic RLVR on the UHR RS benchmark. Our controlled studies yield a counter-intuitive finding: high-quality Earth-science text-only QA is a primary driver of UHR visual reasoning gains. Despite lacking images, domain-specific text injects the concepts, mechanistic explanations, and decision rules necessary to guide visual evidence retrieval. Based on this, we propose a staged knowledge injection recipe: (1) cold-starting with scalable, knowledge-graph-verified Earth-science text QA to instill reasoning structures; and (2) "pre-warming" on the same hard UHR image–text examples during SFT to stabilize and amplify subsequent tool-based RL. This approach achieves a 60.40% Pass@1 on XLRS-Bench, significantly outperforming larger general-purpose models (e.g., GPT-5.2, Gemini 3.0 Pro, Intern-S1) and establishing a new state-of-the-art.

[1]College of Computer Science and Technology, National University of Defense Technology, China [2]Shanghai Artificial Intelligence Laboratory, China [3]Tsinghua University, China [4]University of the Chinese Academy of Sciences, China [5]Sichuan University, China [6]School of Computer Science, Wuhan University, China [7]Zhongguancun Academy, China [8]Chinese Academy of Science, China [9]Renmin University of China, China. Correspondence to: Yulin Wang, Jing Zhang, Wenlong Zhang.

*Proceedings of the 43rd International Conference on Machine Learning*, Seoul, South Korea. PMLR 306, 2026. Copyright 2026 by the author(s).

## 1. Introduction

Ultra-high-resolution (UHR) remote sensing (RS) imagery greatly increases observable detail for Earth science (Wang et al., 2025b). Despite strong progress in both general and remote-sensing MLLMs, current models still struggle in UHR settings where visual evidence acquisition is crucial: localizing tiny task-relevant regions in massive pixel spaces, choosing the right scale, and aligning local cues with semantics. Recent work broadly categorizes post-training into two dominant approaches: cold-start supervised fine-tuning (SFT) (Ouyang et al., 2022b) and reinforcement learning with verifiable rewards (RLVR) (Yue et al., 2025). Based on the RLVR, Agentic RLVR enables active visual exploration by interleaving textual reasoning with acquired visual evidence (OpenAI, 2025b; Yue et al., 2025). While recent work has explored the synergy between SFT and RLVR for MLLMs (Chen et al., 2025a;b) and their impact on reasoning capacity boundaries, the respective roles and interactions of SFT, RLVR, and Agentic RLVR under a shared pretrained model and evaluation setup remain poorly understood in UHR RS scenarios.

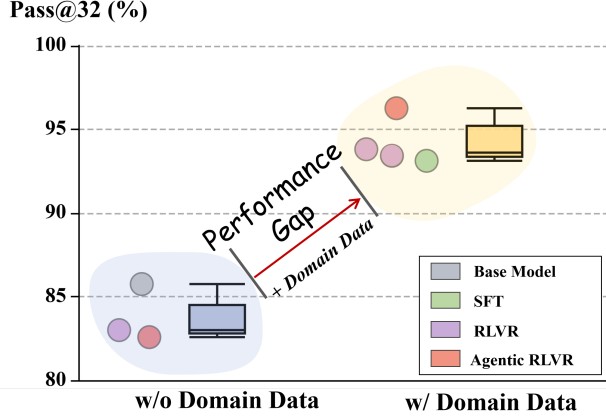

**Pass@32 (%)**

*Figure 2.* **The impact of domain data on performance across different training methods.**

In this paper, we seek to address this limitation. To make the comparison diagnostic, we decompose performance under a fixed inference budget using pass@k curves (Yue et al., 2025). Pass@32 measures the reasoning bound-

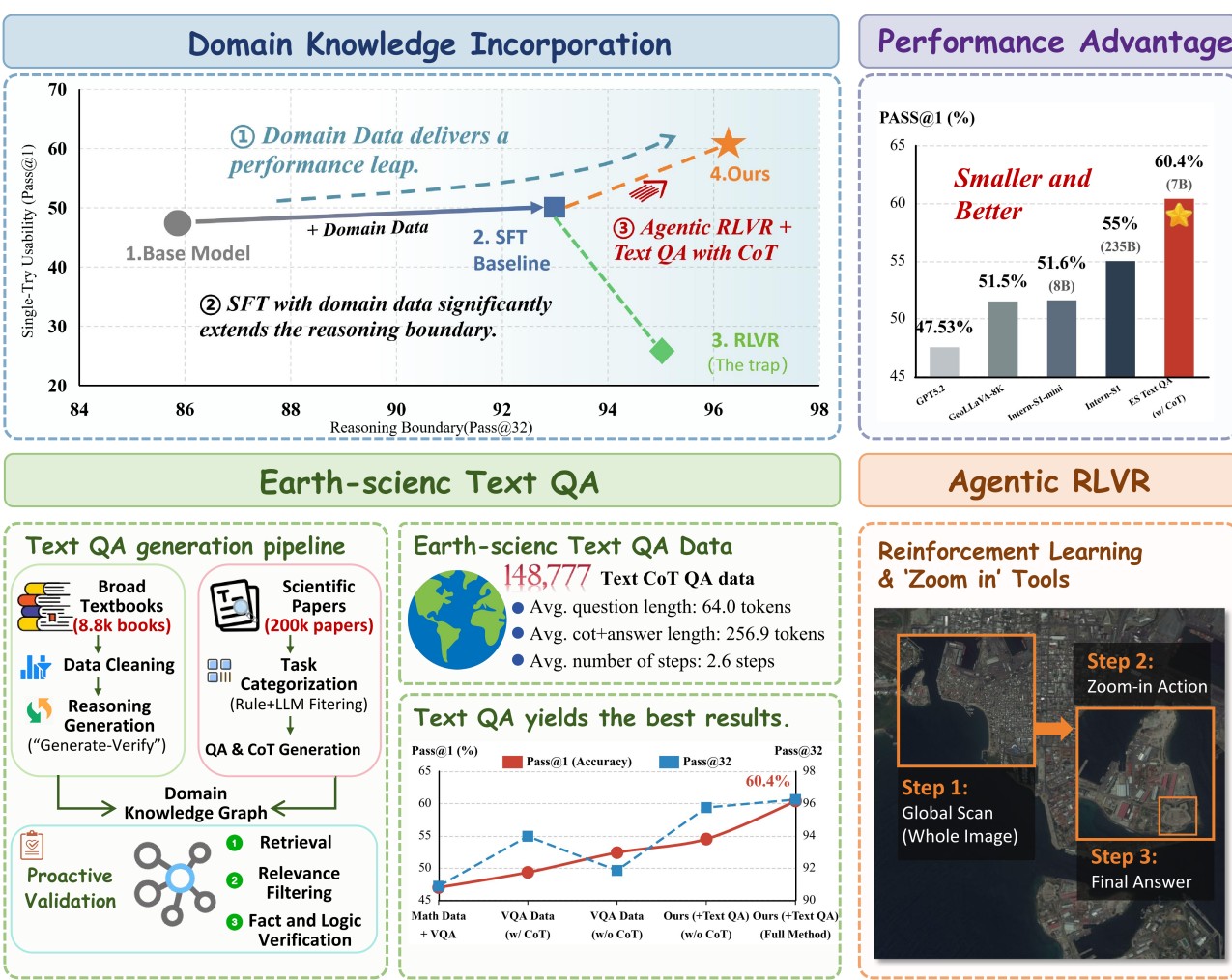

*Figure 1.* We investigate the interplay between post-training paradigms and found that Earth-science text-only QA is a primary driver of UHR visual reasoning gains. Finally, Agentic RLVR, trained on our datasets with our method, significantly outperforms existing MLLMs on UHR RS tasks.

ary—whether at least one correct reasoning trajectory is found among up to 32 samples. On XLRS-Bench (Wang et al., 2025c) which is a widely used UHR RS benchmark that features among the highest image resolutions, we compare three post-training paradigms under controlled settings: SFT, RLVR, and Agentic RLVR that adds image interaction (using zoom-in as the only tool). Figure 2 yield a training design insights: **Reasoning boundary is highly sensitive to high-quality domain data.** Remote-sensing knowledge supervision systematically increases pass@32 within each training method, indicating that reasoning boundary is driven mainly by domain-prior coverage rather than the post-training paradigm itself. Motivated by these findings, our central research question is:

> *How can we effectively inject domain knowledge to improve both the reasoning boundary (pass@32) and the average performance (pass@1) in UHR RS scenarios.*

Further analysis in Sec. 3 yields three critical observations:

(1) **Surprisingly, high-quality Earth-science text-only QA is a major driver of UHR RS reasoning.** Despite lacking images, it encodes domain concepts, mechanisms, and rules that strengthen the model's domain knowledge and even improve visual evidence retrieval during Agentic RLVR, resulting in consistently improving both the reasoning boundary and the average performance.

(2) **UHR RS VQA requires SFT warm-up before RL, and is most effective when paired with text priors.** UHR VQA data is hard to learn in RLVR due to tiny regions or targets in massive pixel spaces, but cold-start SFT warm-up on the same hard samples mitigates this. However, VQA-only warm-up remains limited; the strongest gains are obtained when UHR VQA warm-up is combined with Earth-science text-only QA in cold-start SFT, which provides complementary domain priors that better support downstream exploration and generalization.

(3) **Agentic RLVR becomes effective only under adequate**

**domain supervision.** In the absence of sufficient domain priors, standard RLVR can be unstable and may reduce average performance. With strong text cold-start (and staged VQA warm-up), zoom-in–enabled Agentic RLVR can act as a second-stage optimizer that refines evidence-seeking policies and yields additional gains over standard RL.

Driven by these insights, we propose a unified solution spanning data construction and training. For data, we build a scalable pipeline to generate knowledge-intensive Earth-science text-only QA: two fully automated workflows produce exercise-style and literature-style QA, and an Earth-science knowledge graph filters and controls domain relevance. This end-to-end pipeline reliably yields large-scale, domain-specialized text QA, whose quality and scaling performance were validated in Sec. 5.2. For training, we introduce hard-example pre-warming: we first use these challenging UHR image–text samples for cold-start SFT and then reuse them during Agentic RLVR, initializing spatial/task representations and shifting RL from blind evidence-path exploration to refining zoom-in policies.

Overall, we make the following contributions:

(1) **Mechanistic insights.** We systematically compare SFT, RLVR, and Agentic RLVR in UHR RS settings. We show that the reasoning boundary is primarily governed by domain-prior coverage and further discover that Earth-science text-only QA is a major driver of UHR RS reasoning

(2) **Pipeline of Earth-science text-only QA data.** We develop an automated data-construction pipeline and an Earth-science knowledge graph for quality control, enabling scalable domain supervision with verifiable quality.

(3) **Staged knowledge injection recipe for training.** We propose a cold-starting with Earth-science text QA to instill reasoning structures and a hard-example pre-warming training strategy for VQA RS data.

## 2. Preliminaries

In this section, we briefly describe the pass@k metrics and the datasets used in our experiments. We provide the baseline algorithms for RLVR and Agentic RL in the Supp. A. Note that, given the requirements of UHR settings, we adopt Deepeyes (Zheng et al., 2025)—the zoom-in–enabled agentic RL framework—as our Agentic RL baseline.

**Pass@k Metrics.** The pass@k metric, extended from code generation to all verifiable-reward tasks, measures the fraction of problems solved within $k$ trials. Following previous works (Chen, 2021), for each problem $x_i$ in the evaluation set $\mathcal{D}$ we draw $n$ samples ($n \geq k$) and count correct samples

| Base Model | SFT Data | RLVR Method | RLVR Data | Pass@1 | Pass@32 |
|---|---|---|---|---|---|
| *General Setting (No Domain Data)* | | | | | |
| QwenVL2.5 | × | × | × | 47.36 | 85.75 |
| QwenVL2.5 | × | GRPO | △ | 36.63 | 83.72 |
| QwenVL2.5 | × | GRPO w/ tools | △ | *50.01* | 82.58 |
| *+ Domain-Specific Data* | | | | | |
| QwenVL2.5 | ■ | × | × | 48.26 | 93.11 |
| QwenVL2.5 | ■ | GRPO | △+■ | 25.31 | 95.03 |
| QwenVL2.5 | ■ | GRPO w/ tools | △+■ | *52.39* | 91.85 |
| QwenVL2.5 | ♦+■ | GRPO w/ tools | △+■ | **60.40** | **96.25** |

*Table 1.* **Rigorous ablation studies for the effect of post-training method.** Data icons: × none, ♦ Domain data (our constructed Earth-Science QA pairs), ■ Domain data (SuperRS-VQA), △ General data (DeepEyes-47K).

$c_i$. We then define

$$\text{Pass@k} := \begin{cases} \frac{1}{|\mathcal{D}|} \sum_{x_i \in \mathcal{D}} \frac{c_i}{n} & \text{if } k = 1, \\ \mathbb{E}_{x_i \sim \mathcal{D}} \left[ 1 - \frac{\binom{n-c_i}{k}}{\binom{n}{k}} \right] & \text{if } k > 1, \end{cases} \quad (1)$$

where the $k = 1$ branch corresponds to macro-averaged accuracy. We adopt macro accuracy for $k = 1$ to align with GeoLLaVA-8K(Wang et al., 2025b) and because it better reflects single-shot usability, while the $k > 1$ branch gives an unbiased, low-variance estimate of pass@k for all $k \leq n$.

**Datasets. (1) SuperRS-VQA (Wang et al., 2025b).** It has 12,228 ultra-high-resolution VQA samples (avg. 8376 × 8378, 13 sub-tasks). The tiny targets and dense spatial layouts make it the primary RS set for training. **(2) DeepEyes-47K (Zheng et al., 2025).** It include 47K verifiable-reward samples aggregated from V*, ArxivQA, ThinkLite-VL, etc., to inject diverse reasoning patterns that complement RS data and steady RLVR/Agentic RLVR optimization. **(3) XLRS-Bench (Wang et al., 2025c) .** As a UHR RS benchmark with the highest resolutions in RS benchmark, XLRS-Bench is used to compare SFT, RLVR, and Agentic RLVR under a fixed sampling budget, reporting both pass@1 (average performance) and pass@32 (reasoning boundary). **(4) Earth-Science text QA pairs.** Our 148,777-text CoT corpus supplies Earth-science concepts and rules without images. It serves as cold-start SFT supervision. Detailed pipeline and quality control are described in Section 4.

## 3. Effects of Knowledge Incorporation

In this section, we investigate in greater detail the interplay among post-training paradigms by comparing cold-start SFT, RLVR, and Agentic RLVR on XLRS-Bench. We address three sub-questions: which post-training paradigm best suits UHR RS scenarios., when to incorporate domain knowledge, and how the modality of domain knowledge affects performance. Through controlled ablations and pass@k analysis, we derive reproducible findings, yielding practical training and data-usage guidelines for UHR RS MLLMs.

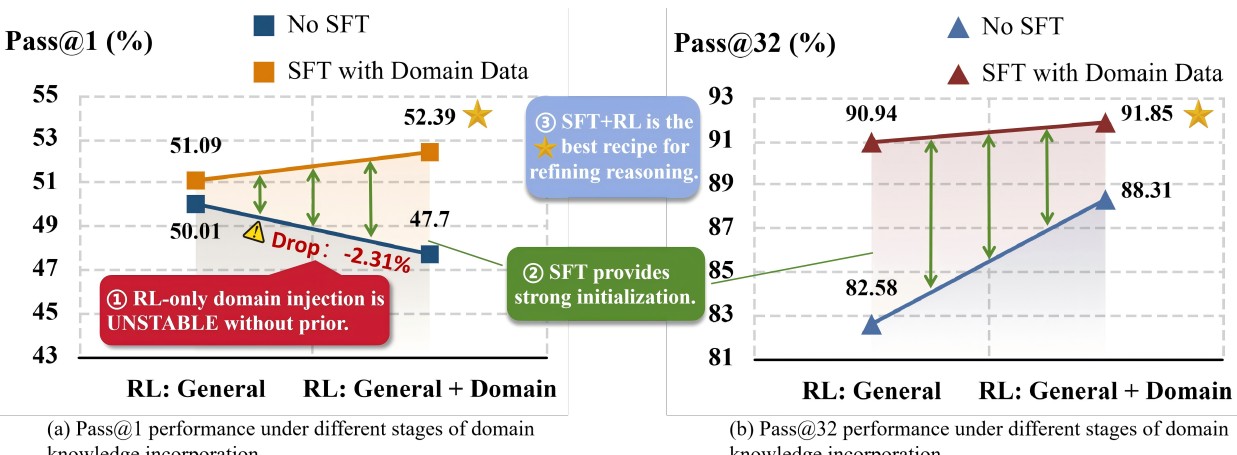

(a) Pass@1 performance under different stages of domain knowledge incorporation.

(b) Pass@32 performance under different stages of domain knowledge incorporation.

*Figure 3.* **The impact of cold-start SFT and RL-stage domain knowledge injection on Pass@1 and Pass@32 in Agentic RLVR.** We only use the VQA data (SuperRs-VQA) as the *Domain Data*. The findings of ① ② ③ are detailed in the main text. Cold-start SFT with domain data yields consistent improvements in both average performance (Pass@1) and reasoning boundary (Pass@32), whereas incorporating domain knowledge only during RL produces smaller or less stable gains, highlighting the importance of staged image-text knowledge incorporation.

## 3.1. Effect of post-training method

**Training Setups and Data.** We use a normal GRPO and a GRPO variant with a zoom-in tool (DeepEyes (Zheng et al., 2025)) as the baseline and keep its general-domain RL data fixed (DeepEyes-47K (Zheng et al., 2025)) across all settings. For domain knowledge, we use SuperRS-VQA (Wang et al., 2025b) and our Earth-science text QA data. Unless noted otherwise, all runs follow the same schedule: 1 SFT epoch with LLaMA-Factory, then 80 RLVR steps. Table 1 shows the detailed results.

**Zoom-in–enabled Agentic RLVR yields more stable average performance gains.** RLVR does not consistently beat SFT on average performance (pass@1) and can even degrade performance without explicit visual evidence-acquisition actions. By contrast, zoom-in–enabled Agentic RLVR yields more stable pass@1 gains and incorporate domain data into single-shot reasoning success more effectively.

**Reasoning boundary is highly sensitive to high-quality domain data, rather than the post-training method.** Remote-sensing knowledge supervision systematically increases pass@32 within each training method, indicating that reasoning boundary is driven mainly by domain-prior coverage rather than the post-training paradigm itself. Notably, Agentic RLVR's gains from VQA-style domain data manifest primarily in average performance (pass@1), with limited improvement in the reasoning boundary (pass@32). This limitation is substantially mitigated once Earth-science text-only QA is introduced, leading to a pronounced increase in pass@32.

---

> **Takeaway 1**
>
> Agentic RLVR delivers strong average performance and effectively leverages domain knowledge to expand the reasoning boundary, making it well suited for UHR RS scenarios.

## 3.2. Effects of training stage of Incorporating Domain Knowledge (SFT vs. RLVR)

Prior results show that incorporating domain knowledge consistently improves the reasoning boundary (pass@32) and Agentic RLVR yields more stable average performance gains. Under Agentic RLVR, however, we need to directly assess how incorporating domain data at different stages impacts both pass@1 and pass@32. We follow the baseline in the above section. For domain knowledge, we use SuperRS-VQA (Wang et al., 2025b) and train it either during cold-start SFT or during tool-augmented GRPO training. The results are shown in the Fig. 3.

① **Incorporating domain knowledge does not necessarily improve pass@1.** With QwenVL2.5 fixed, we keep general-domain RL data (DeepEyes-47K) and all training hyper-parameters unchanged. We only vary the stage at which SuperRS-VQA is introduced: none, RL-only, SFT-only (cold start), or SFT+RL. The results in the Fig. 3 (a) show that domain data effect on average performance (pass@1) is unstable. We attribute this to the fact that tool-using Agentic RLVR need to learn a tool-conditioned visual evidence reasoning path, not just the final answer. Thus, adding domain data alone in RLVR stage is insufficient: coordinated SFT–RL integration is required to improve pass@1 and pass@32.

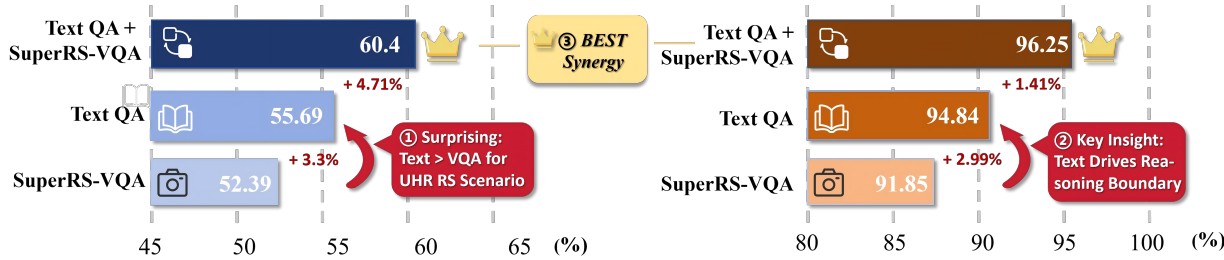

**Text Before Vision**

(a) Pass@1 under different domain knowledge modalities at cold-start.

(b) Pass@32 under different domain knowledge modalities at cold-start.

*Figure 4.* **Effects of domain knowledge modality and injection stage on Agentic RLVR performance.**

② **Cold-start SFT is substantially more effective than RL-stage injection for incorporating domain knowledge.** We directly compare two settings: training SuperRS-VQA during cold-start SFT versus training it during the RL stage. The results in the Fig. 3 show that training SuperRS-VQA in SFT yields consistent gains over RL-only incorporation on both pass@1 and pass@32. We attribute this to SuperRS-VQA's high difficulty and specialization in UHR RS scenarios: introducing it only during RL makes it hard to discover effective reasoning path, whereas SFT-based learning instills robust domain concepts and visual patterns, resulting in better performance in both pass@1 and pass@32.

③ **"Pre-warming" on the same hard UHR image–text examples during SFT to stabilize and amplify subsequent tool-based RL.** We therefore add a two-stage training setting: cold-start SFT on SuperRS-VQA followed by RLVR on the same data. This recipe clearly outperforms using SuperRS-VQA only in SFT or only in RLVR. We posit that SFT provides an initial representation of UHR spatial structure and task patterns, allowing RLVR to focus on refining multi-step reasoning and tool-use policies rather than exploring from a near-random initialization, thus breaking prior performance bottlenecks.

> **Takeaway 2**
>
> For UHR RS image–text expertise, a two-stage strategy which incorporating challenging VQA domain data in both SFT and RL often delivers the most consistent gains.

### 3.3. Effects of Domain Knowledge Modality (Image–Text vs. Text-Only)

In the previous section, we showed that jointly learning the challenging, high-value RS VQA data during both SFT and RL can substantially improve performance on pass@1 and pass@32. This naturally raises a new question: **whether domain knowledge must come in VQA form?** Earth science provides rich text sources—textbooks and papers—that offer expertise comparable to VQA. Thus, we investigate whether text-only QA can also instill domain concepts, phys-

ical processes, and diagnostic reasoning.

**Training Setups and Data.** We use the same GRPO baseline with a zoom-in tool. Beyond SuperRS-VQA, we introduce a large corpus of text-only Earth-science QA with chain-of-thought; its construction and scaling are detailed in Section 4. All other settings match the previous subsection.

① **Text-only data as a cold-start SFT data improves pass@1 more effectively than image–text data.** Although UHR RS VQA benchmarks are highly image-centric and challenging, we find the opposite of the common intuition: incorporating text-only domain knowledge yields larger pass@1 gains than incorporating image–text data during cold-start training. This mirrors general-domain observations (Chen et al., 2025a) that text-only initialization improves both text and multimodal reasoning, while multimodal-only cold-start SFT helps less. In our UHR RS setting, text-only SFT followed by tool-augmented GRPO achieves higher pass@1 than multimodal SFT initialization. We attribute this to the dense Earth-science expertise in text and the language–vision alignment of MLLMs, which enables effective transfer text knowledge to UHR perception and reasoning.

② **Text-only cold-start training yields a substantial improvement in the reasoning boundary (pass@32).** As shown in Fig. 2, boundary gains are primarily driven by incorporating high-quality knowledge. In our new experiments, we find that text data contains dense Earth-science expertise and is often more specialized than image–text VQA data. Consistently, under pass@32, cold-start SFT with text-only data significantly outperforms cold-start SFT with VQA data.

③ **Synergy between text and image–text data.** Prior work suggests that high-quality textual reasoning trajectories can transfer to multimodal tasks even without visual inputs via language–vision alignment in VLMs (Chen et al., 2025b), our results suggest that such transfer is not fully sufficient for UHR remote sensing when text-only data is used in isolation. Combining text with image–text data during cold-start SFT better couples textual reasoning with UHR perception,

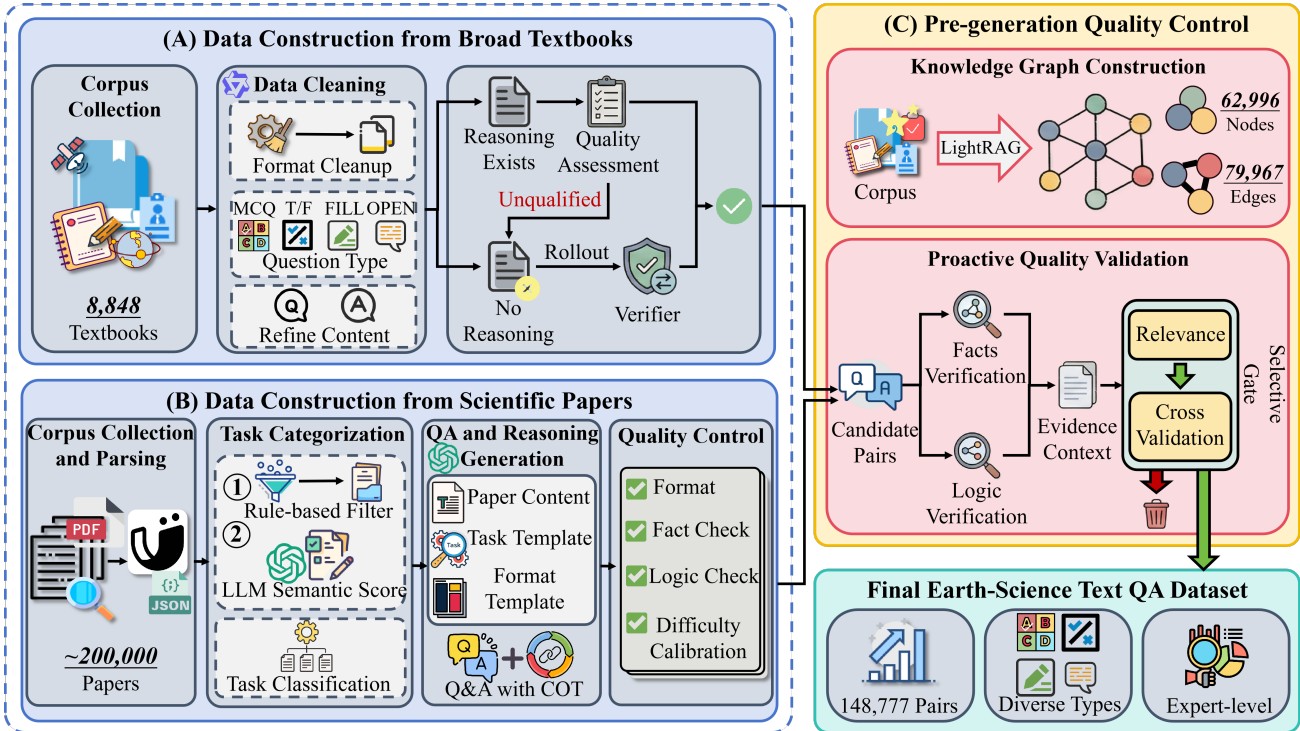

*Figure 5.* **Automated pipeline for Earth-science text QA generation. Panel A** shows textbook-based construction that produces candidate exercise-style QA grounded in foundational concepts through corpus collection, cleaning, normalization, and reasoning refinement. **Panel B** shows paper-based construction that produces candidate literature-style QA targeting frontier topics and complex reasoning through paper parsing, task categorization, and template-guided generation with multi-stage checks. **Panel C** builds a textbook-derived knowledge graph from the cleaned textbook corpus and uses it to screen and validate candidates from Panels A and B for domain relevance as well as factual and logical consistency.

consistently improving both pass@1 and pass@32.

> **Takeaway 3**
>
> High-quality text-only cold-start training boosts both pass@1 and pass@32; combining it with RS VQA at cold start further strengthens and stabilizes these gains.

## 4. An Automated Pipeline for Earth-Science Text QA Generation

Our earlier experiments indicate that high-quality Earth-science text-only QA is a key driver of UHR remote-sensing gains, but scaling such data is nontrivial: Earth-science knowledge is broad and specialized, terminology is dense, expertise is fragmented, and relevant texts are dispersed with limited standardization. To address this, we develop the fully automated two-stage pipeline in Fig. 5. (1) Data construction: we design end-to-end workflows for textbooks and scientific papers to generate exercise-style and literature-style QA, covering both foundational knowledge and frontier reasoning. (2) Quality control: we build an Earth-science knowledge graph to calibrate domain relevance for QA produced from both sources. This pipeline reliably synthesizes large-scale, domain-specialized text QA.

| Statistic | Value |
|---|---|
| Total QA pairs | 148,777 |
| *Overall Statistics* | |
| - Avg. question length | 64.0 |
| - Avg. answer length | 256.9 |
| - Max question length | 1,287 |
| - Max answer length | 4,451 |
| - Question types | MCQ/Fill/TF/Free |
| - Type ratio (%) | 24/7/4/65 |
| *Reasoning Strength* | |
| - Avg. reasoning steps | 2.6 |
| - Max reasoning steps | 17 |
| - Avg. reasoning step length | 96.0 |

*Table 2.* **Main statistics of our dataset.** All lengths are measured in tokens. "Answer" refers to the complete response, including both the reasoning process and the final answer. "Type ratio" reports the proportion of each question type: MCQ (Multiple Choice), Fill (Fill-in-the-Blank), TF (True-or-False), and Free (Free-form QA).

**Data Construction from Broad Textbooks** In this stage, we aim to construct a high-quality QA dataset focus on the fundamental knowledge system of Earth science. To achieve this, we designed a four-step automated pipeline as illustrated in Fig. 5. Our pipeline consists of three steps: corpus collection, data cleaning, and reasoning generation with quality control. More details are shown in the Supp. B.

| SFT Data | | RL | pass@1 | pass@32 |
|---|---|---|---|---|
| Various Text Data | SuperRS | Gen+RS | | |
| ES (w/ CoT) | ✓ | ✓ | **60.40** ⎫ | **96.25** ⎫ |
| ES (w/o CoT) | ✓ | ✓ | 54.49 ⎭ -5.91 | 95.75 ⎭ -0.50 |
| ES (w/ CoT) | ✓ | ✓ | **60.40** ⎫ | **96.25** ⎫ |
| Math (w/ CoT) | ✓ | ✓ | 46.98 ⎭ -13.42 | 90.87 ⎭ -5.38 |
| ES (w/ CoT) | ✓ | ✓ | **60.40** ⎫ | **96.25** ⎫ |
| SuperRS-VQA (w/ CoT) | | ✓ | 49.33 ⎭ -11.07 | 93.97 ⎭ -2.28 |

*Table 3.* Ablation study for the effects of CoT in Earth-Science Text QA. **SuperRS Column**: ✓denotes standard SuperRS-VQA.

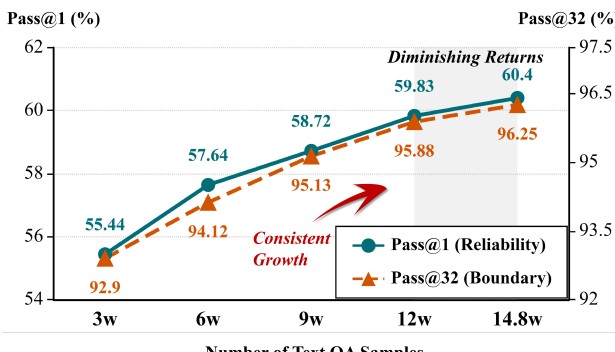

*Figure 6.* **Scaling volume of high-quality Earth-science text QA in cold-start SFT.**

**Data Construction from Broad Scientific Papers** To enhance the precision and complex reasoning capabilities of existing textbook-based knowledge corpora, we developed an automated pipeline for constructing a QA dataset from a broad range of papers, as illustrated in Fig. 5. More implementation details are provided in the Supp. B.

**Pre-generation Quality Control with Large-scale Knowledge Graph** While the previously introduced automated pipeline scales effectively, its reliance on generative models can introduce noise or hallucinations unrelated to core Earth science knowledge. To proactively ensure domain purity and factual accuracy, we implement a pre-generation quality control mechanism. This mechanism uses a structured knowledge graph derived from high-quality textbooks as a verification tool, as illustrated in Fig. 5. Details are shown in the Supp. B.

**Statistics.** Upon completion of the automated construction and quality control pipelines, we consolidated the dual-source data from textbook exercises and academic literature into a unified Earth Science QA dataset, where every sample includes a complete reasoning chain. Table 2 summarizes the key statistics of the dataset.

# 5. Discussion

## 5.1. Effects of CoT in Earth-Science Text QA

In this section, we investigate why Earth-science text data, within an Agentic RLVR framework, improves both perception and reasoning in UHR RS settings. Thorough the ablation study in Table 3, we have discovered that earth-science text QA is an effective cold start for UHR RS because it provides domain-grounded procedural scaffolding (from CoT) and broad domain-prior coverage (from QA content), which together reduce exploration over zoom-in evidence paths and increase the reachability of correct evidence–reasoning trajectories. We reuse SuperRS-VQA, the general-domain RL data, and the Earth-science QA from prior sections, and additionally include s1.1-R1 (Chen et al., 2025a), a text-only math dataset of long CoT trajectories widely used in studies of SFT–RL synergy.

**The contribution of CoT in Earth-science text QA mainly lies in providing an executable reasoning structure rather than adding extra domain knowledge.** In Table 3, we find that removing CoT leads to a much larger drop in pass@1 (-5.91) than in pass@32 (-0.50). This asymmetric effect suggests that Earth-science CoT does not primarily expand the set of correct trajectories reachable under a fixed budget (the reasoning boundary); instead, it cold-starts the model with better procedural reasoning, making it easier to organize visual evidence and apply reasoning rules within a single attempt, thereby improving single-shot success (average performance). In summary, CoT in Earth-science text QA provides strong reasoning-structure priors that substantially boost average performance, while the QA content itself supplies domain priors that markedly increase the reasoning boundary.

Moreover, we also report two key findings: **(1) Domain CoT is essential.** CoT in Earth-science text QA cannot be replaced by math problems with CoT. **(2) Text CoT Matters More Than VQA CoT.** Earth science Text-QA CoT gains are not easily replicated by switching to RS VQA-CoT. Detailed Analyses are offered in the Supp. C

## 5.2. Scaling of Earth science QA pairs

We vary the amount of text used for 1 epoch cold-start SFT, always co-training with SuperRS-VQA. We then hold RL fixed, training for 48 steps on the same general-domain RL data plus SuperRS-VQA, and evaluate on XLRS-Bench with pass@1 and pass@32. The results in Fig. 6 show that scaling high-quality Earth-science text QA for cold-start SFT consistently improves pass@1, while pass@32 plateaus as the reasoning boundary nears saturation. Pass@32, as a boundary reasoning ability, starts high and shows clear diminishing returns: under a fixed inference budget, the model already covers many correct trajectories, and further gains likely require additional long-tail domain concepts and discriminative domain rules rather than more in-distribution samples. By contrast, pass@1 increases steadily with data, suggesting that text cold start primarily improves single-

| Method | Parameters | PASS@1 | PASS@32 |
|---|---|---|---|
| *Remote Sensing MLLMs* | | | |
| GeoChat(Kuckreja et al., 2024) | 7B | 22.03 | – |
| ZoomEarth(Liu et al., 2025a) | 3B | 40.20 | – |
| GeoLLaVA-8K(Wang et al., 2025b) | 7B | 51.50 | – |
| *Closed-source MLLMs* | | | |
| Claude 3.7 Sonnet (Anthropic, 2023) | – | 40.5 | – |
| GPT-o3 (OpenAI, 2025b) | – | 43.6 | – |
| GPT-5.2 (OpenAI, 2025a) | – | 47.53 | – |
| Gemini 2.5 Pro (Comanici et al., 2025) | – | 45.2 | – |
| Grok-4 (xAI, 2025) | – | 45.4 | – |
| *Open-source MLLMs* | | | |
| VLM-R[3] (w/ tools)(Jiang et al., 2025) | 7B | 39.10 | – |
| Qwen2.5-VL(Bai et al., 2025c) | 7B | 47.4 | 85.75 |
| GLM-4.1V(Hong et al., 2025) | 9B | 49.8 | – |
| Qwen3-VL-8B(Bai et al., 2025b) | 8B | 50.02 | – |
| Qwen3-VL-235B-A22B(Bai et al., 2025b) | 235B | 51.11 | – |
| Intern-S1-mini(Bai et al., 2025a) | 8B | 51.6 | – |
| Intern-S1(Bai et al., 2025a) | 241B | 55.0 | – |
| Baseline (Zheng et al., 2025) | 7B | 50.01 | 82.58 |
| *+ pre-warming with SuperRS-VQA* | 7B | 52.39 | 91.85 |
| *+ pre-warming + ES Text QA (w/ CoT)* | **7B** | **60.40** | **96.25** |

*Table 4.* **Comparison of our method with state-of-the-art MLLMs on XLRS-Bench.**

trajectory reasoning success. Scaling Earth-science text QA strengthens domain-grounded reasoning templates and mechanistic causal scaffolds, providing a better structural domain prior before zoom-in–enabled RL training.

### 5.3. Comparison with Other Models

To comprehensively evaluate the effectiveness of our dataset and training approach, we compare our method against a wide range of state-of-the-art MLLMs on the UHR RS benchmark. This comparison includes remote sensing specialized models, closed-source commercial models, and open-source general-purpose vision-language models. All models are evaluated on XLRS-Bench using the same evaluation protocol. For closed-source models, we use their official APIs with default settings. For open-source models, we use their publicly released checkpoints and evaluate under the same inference conditions. Our 7B model with Earth-science text QA cold start and agentic RLVR training outperforms much larger general-purpose models.

## 6. Related Work

**Multimodal Large Language Models for Remote Sensing.** RS multimodal large language models (Wang et al., 2025b; Kuckreja et al., 2024; Hu et al., 2025) are typically built upon general-purpose instruction-following MLLMs. These models extend their capabilities to RS scenarios—such as image captioning(Wang et al., 2024; Hu et al., 2025; Sun et al., 2024a; Liu et al., 2025b; Wang et al., 2026), visual

question answering (VQA)(Kuckreja et al., 2024),visual grounding(Guo et al., 2024), and multi-turn dialogue(Zhang et al., 2024a) —through RS-specific instruction fine-tuning. However, in UHR RS scenarios, these models struggle to accurately locate task-relevant fine-grained regions within vast pixel spaces. To address the challenges of UHR environments, existing research has proposed various methodologies: Supervised Fine-Tuning (GeoLLaVA-8K (Wang et al., 2025b), ImageRAG(Zhang et al., 2024b), and RFM (Luo et al., 2025)), Reinforcement Learning (ZoomEarth (Liu et al., 2025a)), and Agentic frameworks that integrate tool invocation for multi-turn interactive evidence acquisition and reasoning (ICoT-Agent (Wang et al., 2025a)). Meanwhile, numerous UHR remote sensing datasets (Wang et al., 2025c; Luo et al., 2025; Liu et al., 2025a) provide diverse tasks for evaluating performance in high-resolution settings. Among these, XLRS-Bench (Wang et al., 2025c) offers extensive range of task types and significantly higher resolutions than other benchmarks.

**SFT and RL Synergy in Multimodal Models** In the realm of general Large Language Models, post-training typically combines SFT with RL-style optimization to achieve preference alignment and capability shaping. Specifically, SFT adapts pre-trained models to task-specific behavioral patterns via supervised learning on curated instruction pairs (Ouyang et al., 2022a). RLVR, conversely, builds upon the pre-trained model, utilizing automatically computable reward signals to further enhance generation quality through reinforcement learning (Cobbe et al., 2021). Whether RLVR genuinely expands reasoning capabilities beyond SFT remains a subject of debate; some view RL primarily as a refiner (Yue et al., 2025; Shao et al., 2025; Yeo et al., 2025; Wu et al., 2025; Wang et al., 2025d; Gu et al., 2025; Ai et al., 2025), while others report substantive gains surpassing SFT (Wen et al., 2025; Yuan et al., 2025; Sun et al., 2025). In the context of Multimodal Large Language Models, several studies have also begun to explore the synergy between RL and SFT (Sun et al., 2024b).

## 7. Conclusion

Agentic RLVR with zoom-in is a natural candidate for UHR RS reasoning, yet our controlled study shows that domain data, not the post-training paradigm alone, dominates the achievable gains in this regime. By decomposing performance into average performance (pass@1) and the fixed-budget reasoning boundary (pass@32), we observe a counter-intuitive but robust pattern: high-quality Earth-science text-only QA is the primary driver of UHR gains, expanding the reasoning boundary while also strengthening average performance even without images. Guided by this diagnosis, we propose a staged "Text-Before-Vision" recipe: text-first cold-start SFT to instill domain concepts,

mechanistic reasoning, and decision rules, plus SFT pre-warming on the same UHR hard examples that will later be optimized, so that tool-based RL shifts from blind exploration in massive pixel spaces to second-stage refinement of evidence-seeking policies. With this data-centric strategy, our Agentic RLVR system reaches 60.40 pass@1, surpassing the 55.0 pass@1 reported by the 241B science-specialized Intern-S1, while also improving pass@32 by 13.67 points over the Agentic RLVR baseline, yielding a practical and reproducible recipe for jointly improving both average performance and reasoning boundary on XLRS-Bench.

## Acknowledgements

This work was supported in part by the New Generation Artificial Intelligence-National Science and Technology Major Project (No. 2025ZD0123602), the National Natural Science Foundation of China (No. 624B2109, No.62376282 and No.62525213), the Fundamental and Interdisciplinary Disciplines Breakthrough Plan of the Ministry of Education of China (No. JYB2025XDXM101), the Science and Technology Innovation Program of Hunan Province (No.2025RC3117) and the Provincial Natural Science Foundation of Hunan 2025JJ10008.

## Impact Statement

This paper presents work whose goal is to advance the field of Machine Learning. There are many potential societal consequences of our work, none which we feel must be specifically highlighted here.

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

# A. Basic Algorithm

## A.1. Reinforcement Learning with Verifiable Rewards

**Verifiable Rewards.** A language model with parameters $\phi$ generates $\mathbf{o} = (o_1, \ldots, o_L)$ for query $q \sim Q$ via policy $P_\phi(\mathbf{o} \mid q)$. A programmatic evaluator $\mathcal{E}$ returns binary correctness $c \in \{0, 1\}$; optional formatting constraints keep reasoning and answer separated. The objective maximizes expected correctness:

$$\mathcal{L}(\phi) = \mathbb{E}_{q \sim Q} \left[ \sum_{\mathbf{o}} P_\phi(\mathbf{o}|q) \cdot \mathcal{E}(q, \mathbf{o}) \right], \tag{2}$$

where the expectation integrates over queries from $Q$, and the summation aggregates over all possible output sequences weighted by their generation probabilities under the current policy $P_\phi$.

**RLVR Algorithms.** Proximal Policy Optimization (PPO) is adopted; the clipped surrogate constrains the probability ratio $\rho_i(\phi) = P_\phi(o_i|q, \mathbf{o}_{<i})/P_{\phi_{\text{ref}}}(o_i|q, \mathbf{o}_{<i})$ to stabilize updates:

$$\mathcal{U}(\phi) = \mathbb{E} \left[ \min \left( \rho_i(\phi) \cdot \delta_i, \text{clip} \left( \rho_i(\phi), 1 - \delta, 1 + \delta \right) \cdot \delta_i \right) \right], \tag{3}$$

where $\delta_i$ is the advantage from value network $U_\psi$ and $\delta$ is a clipping hyperparameter; an optional KL penalty further limits deviation from the reference policy.

## A.2. Agentic Reinforcement Learning

**Rollout Formulation.** Agentic RL augments text-only CoT with observation tokens from external tools; the state at step $k$ is the interleaved history of model text $u_k$ and observations $v_k$:

$$z_k = \left\{ (u_0, v_0), (u_1, v_1), \ldots, (u_k, v_k) \right\}. \tag{4}$$

Observation tokens are masked out from the language-model loss.

**Reward Design.** Reward combines correctness, formatting, and a conditional tool bonus applied only when the answer is correct:

$$G(\gamma) = S_{\text{ok}}(\gamma) + S_{\text{fmt}}(\gamma) + \mathbf{1}[S_{\text{ok}}(\gamma) = 1] \cdot S_{\text{tool}}(\gamma), \tag{5}$$

where $\mathbf{1}[\cdot]$ is the indicator function.

**Optimization.** Group Relative Policy Optimization (GRPO)(Shao et al., 2024) is used; token-level masking restricts gradients to model-generated tokens, treating tool observations purely as conditioning.

# B. Data pipeline

## B.1. Data Construction from Broad Textbooks

**Data Construction from Broad Textbooks** In this stage, we aim to construct a high-quality QA dataset focus on the fundamental knowledge system of Earth science. To achieve this, we designed a four-step automated pipeline as illustrated in Fig. 5. Our pipeline consists of three steps: corpus collection, data cleaning, and reasoning generation with quality control.

**Corpus collection.** We compile a collection of 8,848 high-quality textbooks and accompanying exercises in Earth science and remote sensing. The selection of data sources follows a dual criteria approach. For breadth, we ensure across the domain, encompassing the spectrum from remote sensing fundamentals to meteorological attribution. For quality, professionally reviewed and published materials are prioritized to ensure terminological and definitional standardization.

**Data cleaning.** To ensure compatibility for subsequent analysis, multi-source data are first transformed and standardized into a unified format. Specifically, we apply a hybrid strategy combining rule-based and model-based methods to remove noise and inconsistencies from the raw corpus. Following this, LLMs are employed to automatically classify question types and map them to predefined templates, thereby achieving structural normalization across the entire dataset. Then, to further enhance content quality and consistency, we apply another round of LLMs to perform deep validation and rewriting of each QA pair, preventing semantic-level noise like mismatched answers, cue leakage within questions, or ambiguous phrasing.

*Table 5.* **Task Hierarchy and Definitions.** A two-level hierarchy comprising three cognitive dimensions and six Level-2 tasks to cover capabilities from basic understanding to methodological application.

| Level-2 Task | Definition | Example |
|---|---|---|
| **Facts and Concepts** | | |
| Conceptual Clarification | Explain the scientific meaning, scope, and common misconceptions of a term, metric, or technical product. | "What is NDVI, and why does it saturate in dense vegetation?" |
| Empirical Summarization | Summarize key observations, experimental findings, or data characteristics from the literature. | "How does urban heat island intensity vary across seasons?" |
| **Mechanisms and Relations** | | |
| Mechanistic Explanation | Reveal mechanisms, causal relations, and condition dependence among variables. | "How do increased aerosols affect surface shortwave radiation?" |
| Relational Analysis | Analyze and compare links, differences, and applicability across concepts, methods, or variables. | "Compare Landsat and Sentinel-2 for land-cover mapping (spatial resolution vs. revisit frequency)." |
| **Analysis and Decision** | | |
| Quantitative Estimation | Estimate magnitudes or perform scale analysis under explicit assumptions. | "Estimate peak runoff from rainfall intensity and catchment area, and state key assumptions." |
| Methodological Selection | Choose suitable data, models, or workflows for a goal, with rationale and limitations. | "When should SAR or optical imagery be used for glacier boundary extraction, and why?" |

**Reasoning generation.** Reasoning generation proceeds via a two-track quality control pipeline. For QA pairs with pre-existing reasoning, we apply multi-dimensional quality assessments, retaining and refining only high-scoring samples. Conversely, pairs lacking or containing low-quality reasoning enter a controlled generate-and-verify cycle: a generator model produces reasoning chains, which are then evaluated by an independent verifier for logical consistency and factual accuracy. Each sample is permitted a single revision. Failure to pass verification after rewrite resulted in discarding, ensuring both structural uniformity and factual reliability in the final dataset.

### B.2. Data Construction from Broad Scientific Papers

To enhance the precision and complex reasoning capabilities of existing textbook-based knowledge corpora, we developed an automated pipeline for constructing a QA dataset from a broad range of papers, as illustrated in Fig. 5. **(1) Corpus collection and parsing:** We First collect approximately 200,000 Earth science papers. Then, we use MinerU to parse and convert these papers into structured JSON text with clear separation of title, abstract, and body text. **(2) Task categorization:** We first utilize a two-step screening strategy to ensure domain purity: initial rule-based filtering followed by LLM-based semantic relevance scoring. Then, these papers are automatically categorized into predefined research tasks based on their abstracts. **(3) QA and reasoning generation:** Given paper content, task template and format template, we use GPT-5 to generate QA pairs with diverse question types (*e.g.*, multiple-choice, open-ended, etc.). Each generated QA pair is accompanied by a standard answer and an explicit chain-of-thought. **(4) Quality control:** Finally, a multi-step pipeline is designed to ensure quality of QA pairs through successive validation of format, factual accuracy, and logical rigor, culminating in difficulty calibration to select challenging, high-quality samples.

### B.3. Pre-generation Quality Control with Large-scale Knowledge Graph

While the previously introduced automated pipeline scales effectively, its reliance on generative models can introduce noise or hallucinations unrelated to core Earth science knowledge. To proactively ensure domain purity and factual accuracy, we implement a pre-generation quality control mechanism. This mechanism uses a structured knowledge graph derived from high-quality textbooks as a verification benchmark, as illustrated in Fig. 5. The core idea is to assess and filter potential QA content *before* its final generation, shifting from passive post-hoc filtering to active pre-emptive validation.

**Knowledge graph construction.** The knowledge graph is built from the cleaned textbook corpus described in Sec. B.1. Specifically, we employ LightRAG to first extract key entities and relationships from the text segments, organizing them

into a searchable graph that encapsulates the domain's fundamental knowledge system.

**Proactive quality validation.** For each candidate QA pair $(q, a)$ proposed by the generator, we perform a two-tier retrieval against the built knowledge graph. The first tier retrieves fine-grained evidence for factual entities within the answer, while the second tier retrieves broader contextual passages to validate the logical coherence and reasoning underpinning the answer. This process yields a supporting evidence context $C_q$ for each candidate. Then, the candidate $(q, a)$ is retained only if its content is sufficiently supported by the evidence $C_q$ from the graph, according to the following criteria: (1) Relevance screening. we compute the semantic relevance between $q$ and $C_q$. If $C_q$ is empty or the relevance score falls below a threshold $\tau$, the pair is filtered as out-of-domain or insufficiently grounded. (2) Factual & logical verification. For pairs passing the relevance check, the factual claims and reasoning chains in $a$ are rigorously cross-checked against $C_q$. Answers containing statements that contradict $C_q$ or that cannot be logically inferred from it are discarded as potential hallucinations.

This pre-generation quality control pipeline, anchored by a textbook-derived knowledge graph, ensures that the final QA dataset possesses high domain-specific fidelity and factual reliability, providing a robust foundation for subsequent model instruction.

## C. Detailed Analyses

**CoT in Earth-science text QA cannot be replaced by math problems with CoT.** UHR remote-sensing requires domain-grounded reasoning rather than abstract general proficiency. Prior work has shown that text-only cold starts on math or logic problems can improve general reasoning (Chen et al., 2025a). Accordingly, we test this by swapping Earth-science text QA (with CoT) for text-only math QA (with CoT) while keeping the rest fixed. The substitution degrades both pass@1 and pass@32 (46.98/90.87), weakening average performance and shrinks the reasoning boundary under a fixed budget. We attribute this to the key bottlenecks in UHR RS—visual evidence acquisition and domain-concept understanding—whereas math CoT mainly reinforces symbolic computation whose reasoning states and failure modes mismatch RS evidence discrimination, limiting transfer to executable domain trajectories.

**Earth science Text-QA CoT gains are not easily replicated by switching to RS VQA-CoT.** Since Earth-science text CoT mainly boosts pass@1 with limited effect on pass@32, we test whether its structured priors can be replaced by image–text CoT. We generate CoT for SuperRS-VQA and run controlled comparisons with all other settings fixed. Using SuperRS-VQA (with CoT) alone performs markedly worse than cold-starting with "text QA + SuperRS-VQA (with CoT)" on both pass@1 and pass@32, showing that VQA-CoT does not reproduce the benefits of text CoT. VQA-CoT is typically answer-centric, lacks stepwise evidence binding aligned with zoom-in evidence seeking, and its tight coupling to visual content can bias RL toward suboptimal zoom-in decisions. Overall, image–text CoT alone is an imperfect substitute for the structural effect of text CoT and may even trade off average performance against boundary gains.

*Table 6.* **Earth-Science Text QA Dataset Examples.** Four distinct question types with reference answers explicitly incorporating a chain-of-thought reasoning process.

| Type | Examples (Question & CoT Answer) |
|---|---|
| **Multiple Choice** | **Question:** Which combination of Southern Hemisphere Annular Mode (SAM) and Southern Oscillation (SO) polarities best explains the observed spatial pattern of warming in the Antarctic Peninsula and peripheral seas, cooling over East Antarctica, and ice edge retreat in the Pacific sector of the Southern Ocean during 1982–1998? *A) Positive SAM and positive SO     B) Positive SAM and negative SO     C) Negative SAM and positive SO     D) Negative SAM and negative SO* 

 **Answer:** Step 1: Positive SAM is associated with cooling over East Antarctica and warming in the Peninsula. 
 Step 2: Negative SO (El Niño) is linked to ice edge retreat in the Pacific sector. 
 Step 3: The combined drift toward positive SAM and negative SO creates the specific observed spatial pattern. 

 **Therefore, the final answer is:** B |
| **Fill-in-the-blank** | **Question:** A soil sample with a cross-sectional area of 0.01 m² and a length of 0.1 m was tested using a steady upward flow constant head permeameter. If 0.5 liters of water flowed through the sample over 120 seconds under a hydraulic head difference of 0.05 m, calculate the saturated hydraulic conductivity (K) in m/s using the formula $K = (V \times L) / (A \times t \times h)$. Fill in the blank: The calculated saturated hydraulic conductivity is _____ m/s. 

 **Answer:** Step 1: Convert the volume from liters to cubic meters: 0.5 L = 0.0005 m³ (since 1 L = 0.001 m³). 
 Step 2: Plug the values into the formula: $K = (0.0005 \times 0.1) / (0.01 \times 120 \times 0.05) = 0.00005 / 0.06 = 0.000833$ m/s. 

 **Therefore, the final answer is:** 0.000833 |
| **True / False** | **Question:** Artesian wells are characterized by water levels that are always above the surface, allowing water to flow out without the need for pumping. True or False? 

 **Answer:** Step 1: Artesian wells tap into a confined aquifer where water is under pressure due to impermeable layers (aquicludes). This pressure causes the water level in a well to rise above the top of the aquifer itself. 
 Step 2: While this pressure can drive water to the surface (flowing artesian well), the potentiometric surface (the level to which water will rise) is not always above the ground elevation. It depends on the specific recharge altitude and geological conditions. 
 Step 3: Therefore, while all artesian wells have water levels above the aquifer's top, they do not "always" have water levels above the land surface. 

 **Therefore, the final answer is:** False |
| **Open-ended** | **Question:** How does the microwave emission model based on Wilheit's (1978) layered dielectric approach, which incorporates solutions to Maxwell's equations and boundary conditions, contribute to calculating brightness temperature from a soil surface, and what specific assumptions or parameters are required for this process? 

 **Answer:** Step 1: The model treats soil as a layered dielectric system, requiring Maxwell's equations to calculate electric fields at each layer's boundary, which determines how electromagnetic waves propagate through the medium. 
 Step 2: Energy fluxes and fractional absorption in each layer are derived from these electric fields, linking electromagnetic properties to thermal radiation characteristics of the soil. 
 Step 3: Brightness temperature contributions from each layer depend on their temperature and dielectric constant, with the model assuming a simplified temperature profile and requiring layer thickness data. 

 **Therefore, the final answer is:** The model calculates brightness temperature by first solving Maxwell's equations to determine electric fields in each soil layer, which are then used to compute energy fluxes and fractional absorption. Each layer's contribution to brightness temperature is derived from its temperature and dielectric properties, with assumptions of constant initial temperature (prior to irrigation) and layered soil structure. Key parameters include dielectric constant, temperature distribution with depth, and layer thicknesses. |

