# OpenReview forum: "Text Before Vision: Staged Knowledge Injection Matters for Agentic RLVR in Ultra-High-Resolution Remote Sensing Understanding"
_ICML.cc/2026/Conference — ICML 2026 regular_

### Official Review · Reviewer_9bTn · 2026-03-06

**Soundness:** 4
**Presentation:** 4
**Significance:** 3
**Originality:** 4
**Overall Recommendation:** 6
**Confidence:** 5

**Summary:**

The authors systematically study the interactions among post-training paradigms on UHR remote-sensing benchmarks, comparing cold-start supervised fine-tuning (SFT), RLVR, and agentic RLVR. Beyond the standard pass@1 metric (average performance), they also conduct a comprehensive comparison of pass@32 as a proxy for the reasoning boundary. Through detailed controlled experiments, they reach a counterintuitive conclusion: high-quality Earth-science text-only QA is the primary driver of improved UHR visual reasoning. Despite containing no images, domain-specific text provides the concepts, mechanistic explanations, and decision rules needed to guide visual evidence retrieval. Motivated by this finding, the authors propose a staged knowledge-injection strategy. They also release an end-to-end pipeline for constructing Earth-science text-only QA data to facilitate community scaling, including a domain knowledge graph for quality control in addition to standard data-generation steps. Finally, they demonstrate strong performance on UHR remote-sensing evaluation benchmarks.

**Compliance With Llm Reviewing Policy:**

Affirmed.

**Final Justification:**

The authors’ rebuttal and additional data-scaling experiments have satisfactorily addressed my main concern. I am therefore raising my score, as I believe this work has the potential to make a broad impact on the community.

**Key Questions For Authors:**

A few minor details:

1. In Figure 1, there are two circles in the top-right corner labeled RLVR. Do they represent different variants, or is this a typo?

2. Figure 4 shows the impact of domain-knowledge modality on training outcomes. The caption states that it compares different cold-start datasets. After cold-starting with different data, did each model undergo the same agentic RL training? If so, it would be helpful to state this explicitly in the caption.

3. Regarding data scaling, please refer to the Weaknesses section.

**Limitations:**

yes

**Strengths And Weaknesses:**

**Strengths：**

1. The paper is the first to systematically investigate the performance of different post-training paradigms in UHR remote-sensing settings. Through extensive and rigorous ablation studies, the authors analyze the respective roles and interactions of SFT, RLVR, and agentic RLVR under a shared pretrained model and evaluation protocol, and quantify their impact on the reasoning boundary. This offers highly actionable insights for addressing UHR challenges in remote sensing.

2. Building on this comprehensive analysis, the authors propose three constructive training recommendations. Most notably, controlled experiments yield a counterintuitive finding: high-quality Earth-science text-only QA is the primary driver of improved UHR visual reasoning. The authors argue that, despite the absence of images, domain-specific text provides the concepts, mechanistic explanations, and decision rules needed to guide visual evidence retrieval.

3. The authors contribute a complete pipeline for large-scale generation of high-quality Earth-science data, along with a domain knowledge graph for expert-level quality control. This is valuable for the remote-sensing community and can facilitate further scaling of Earth-science text resources.

4. The discussion on the role of CoT in Earth-science text QA is particularly valuable. Through careful ablations, the authors report three findings and emphasize that CoT primarily contributes executable reasoning structure rather than injecting additional domain knowledge.

**Weaknesses:**

I am particularly interested in the scaling analysis in Section 5.2 and would like to discuss it further with the authors. Figure 6 suggests that increasing the amount of high-quality Earth-science text QA yields consistently positive gains. However, pass@32 already exceeds 95%, which appears close to saturation. If the community continues to scale Earth-science text data along the authors’ direction, will a clear bottleneck emerge? My intuition is that pass@32, as a proxy for the reasoning boundary, may plateau, while pass@1, the more standard average-performance metric, could still improve differently. I would therefore be very interested to see how both pass@32 and pass@1 evolve as the text dataset is further scaled. Although expanding the dataset substantially in the short term is challenging, I hope the authors can provide more actionable guidance to the community. If the authors can further scale the data and offer more results into how Earth-science text scaling benefits UHR reasoning under both pass@32 and pass@1, I would consider further raising my score.

**Overall, this paper offers a careful analysis of three post-training paradigms, surfaces the importance of Earth-science text-only QA, and releases a reproducible end-to-end data pipeline. I believe it is worth to accept, it will be a valuable contribution to the community, and my scaling comments in weakness do not detract from its contributions.**

---

> ### Author Rebuttal · Authors · 2026-03-30
>
> We sincerely thank the reviewer for the highly positive assessment of our paper and for the constructive suggestion regarding the scaling analysis.
>
> ## Q1: Figure 1
> Thank you for pointing this out. These two points are not a typo, but they do correspond to two closely related settings that are easy to confuse in the current figure. Specifically, both use the same GRPO-based training paradigm and both include domain-specific remote-sensing data. The difference is only in the training data composition: one uses SuperRS-VQA only, while the other uses DeepEyes-47K + SuperRS-VQA.
>
> That said, we agree that showing both points in this figure is unnecessary. To make the presentation cleaner and avoid confusion, we will remove one of the two points in the revised version and keep only a single representative RLVR point.
>
> ## Q2: Figure 4 caption
> Yes. After varying the cold-start dataset, **all models undergo the same second-stage zoom-in-enabled Agentic RL training**; only the cold-start supervision differs. We will revise and clarify the caption.
>
> ## Q3: Data scaling
> We fully agree that the key question is not only whether high-quality Earth-science text QA helps, but also how its gains split between reasoning-boundary expansion (pass@32) and average performance improvement (pass@1) as the text corpus scales. Following your suggestion, we have extended the scaling study and now report **both pass@1 and pass@32 at finer text-data scales**.
>
> |Earth-science text QA used in SFT|Pass@1|ΔPass@1|Pass@32|ΔPass@32|
> |-|-|-|-|-|
> |148k|60.40|-|96.25|-|
> |168k|60.97|+0.57|96.40|+0.15|
> |200k|61.67|+1.27|96.92|+0.67|
>
> This closely matches our observation in Sec. 5.2: under a fixed RL stage and inference budget, scaling high-quality Earth-science text QA consistently improves pass@1, whereas pass@32 shows clear diminishing returns and gradually plateaus as the reasoning boundary approaches saturation.
>
> Our updated results suggest the following practical guidance:
>
> * When the goal is to improve **average-case usability**, continuing to scale high-quality Earth-science text QA remains effective.
> * When the goal is to push the **reasoning boundary** further after pass@32 is already high, the bottleneck shifts from raw data volume to **coverage quality**, especially long-tail concepts and harder domain rules.
>
> We will incorporate these additional results and the corresponding discussion into the revised paper, and we thanks for motivating this clarification.
> ## Reviewer gvqX: Could the authors also evaluate closed-source models in Table 4 with the same zoom-in tool and sampling budget?
>
> We make fair Pass@1 comparisons with both general open-source models and remote sensing specialized models, including those equipped with zoom-in tools. We clarify this in three points below.
>
> * **On the two Pass@ metrics.** Our SOTA claim of **60.40%** is based strictly on **Pass@1 (average performance)**. This is the standard protocol widely used in remote sensing MLLMs (e.g., ZoomEarth, CVPR 2026; GeoLLaVA-8K, NeurIPS 2025) and general open-source MLLMs (e.g., DeepEyes, ICLR 2026). We report Pass@32 (96.25%) separately only as an analysis metric to probe the theoretical reasoning boundary, not for primary baseline comparison.
>
> * **On the zoom-in tool.** ZoomEarth (CVPR 2026), a remote sensing MLLM using the same zoom-in tool, achieves only **40.20%** Pass@1. VLM-R3 and DeepEyes (ICLR 2026), general-domain models using the same tool, achieve only **39.10%** and **50.01%**, respectively. In contrast, we take DeepEyes—already equipped with the zoom-in tool—as the baseline and, without changing the model architecture, raise performance to **60.40%** through our training strategy and data. We therefore attribute the gain in our SOTA result to training strategy and data, rather than to tool use itself. We notice that your suggestion also includes comparing against closed-source API models augmented with a zoom-in tool. However, as noted in DeepEyes itself (also our baseline), closed-source models are end-to-end systems that do not rely on manually designed tool-calling workflows (see Table 1 of DeepEyes). Accordingly, we follow the prior work such as DeepEyes: comparing against open-source MLLMs equipped with zoom-in tools.
>
> * **On the 32-sample budget.** Although evaluating closed-source models under Pass@32 requires roughly 32 times more tokens than standard evaluation, we are willing to report Pass@32 results for closed-source models as well.
>
> |Method|Pass@1|Pass@32|
> |-|-|-|
> |GPT-5.2|47.53|86.24|
> |Gemini 2.5 Pro|45.20|84.49|
> |Ours|63.15|97.32|
>
> Even closed-source models perform only moderately in remote sensing without high-quality domain-specific training data, further highlighting the need for effective training strategies and post-training data in this field.

---

> > ### Author Rebuttal · Reviewer_9bTn · 2026-04-02
> >
> > Thanks to the author's detailed rebuttal, which has addressed my concern. After reading the other reviewers’ comments and the authors’ rebuttal, I will maintain my score and hope the authors will incorporate the corresponding revisions in the final version.

---

> > > ### Author Response · Authors · 2026-04-06
> > >
> > > Thank you again for your insightful feedback and strong support. As you noted in your initial comments, you would consider raising your score further. If you feel that our additional experiments and clarifications have adequately addressed your concerns, we would be sincerely grateful if this could be reflected in your final evaluation. We also deeply appreciate your highly positive assessment of our work.

---

### Official Review · Reviewer_gvqX · 2026-03-08

**Soundness:** 3
**Presentation:** 3
**Significance:** 2
**Originality:** 2
**Overall Recommendation:** 4
**Confidence:** 3

**Summary:**

This paper investigates the interplay between **cold-start SFT**, **RLVR**, and **Agentic RLVR** for **ultra-high-resolution (UHR) remote sensing** reasoning. Through controlled ablations on XLRS-Bench, the authors observe that: (1) Agentic RLVR yields stable average performance gains and effectively leverages domain knowledge to expand the reasoning boundary; (2) using the same hard VQA samples in both cold-start SFT and RL yields the most consistent gains; and (3) text-only Earth-science QA boosts both pass@1 and pass@32, and combining it with RS VQA at cold start further strengthens these gains.

Based on these findings, the paper proposes a staged **"Text-Before-Vision"** recipe — text-only SFT cold start followed by hard-example pre-warming and Agentic RLVR with a zoom-in tool. The paper also presents an automated pipeline that generates 148K Earth-science text QA pairs from textbooks and scientific papers, with a knowledge-graph-based filter for domain relevance and factual consistency. A 7B model trained with this recipe achieves **60.40% pass@1** on XLRS-Bench, outperforming GPT-5.2, Gemini 2.5 Pro, and Intern-S1 (235B).

**Compliance With Llm Reviewing Policy:**

Affirmed.

**Final Justification:**

I thank the authors for their effort in the rebuttal. I acknowledge this work as one of the first in-depth investigations of SFT + RLVR in the remote sensing domain and therefore increase my score.

**Key Questions For Authors:**

1. Could the authors clarify which findings are specific to UHR RS versus confirmations of known general-domain results [1, 2, 3, 4]?
2. Could the authors provide an ablation of downstream performance with and without KG-based filtering?
3. Do the findings hold on other base models (e.g., Qwen3-VL, InternVL) or other UHR RS benchmarks beyond XLRS-Bench?
4. Could the authors also evaluate closed-source models in Table 4 with the same zoom-in tool and sampling budget for a fair comparison?

**Limitations:**

The authors do not discuss limitations of their work in detail. Please refer to weaknesses section.

**Strengths And Weaknesses:**

### Strengths

1. The paper provides a thorough ablation study that systematically varies one factor at a time (post-training paradigm, knowledge injection stage, data modality) and reports both pass@1 and pass@32, making it easy to understand which design choice contributes what.
2. Decomposing performance into pass@1 and pass@32 provides more nuanced insights than single-metric evaluation — e.g., revealing that domain data primarily expands the reasoning boundary while CoT primarily improves single-shot success.
3. The automated Earth-science text QA generation pipeline with knowledge-graph-based filtering is a reusable contribution for the remote sensing community.
4. The paper is clearly structured and easy to follow.

### Weaknesses

1. (Major) The originality of the work is limited. Most findings are domain-specific validations of results already established in the general domain: Takeaway 1 (Agentic RLVR outperforms standard RLVR) follows from DeepEyes [1]; Takeaway 3 (text-only cold start improves VLM reasoning) is well-known [2, 3, 4]. Takeaway 2 (using hard VQA samples in both SFT and RL) is more specific to UHR RS, but the overall contribution remains incremental. The paper would benefit from more clearly stating what is new to the UHR RS setting versus what is a known property being re-confirmed.
2. The KG-based quality control is the key differentiator of the data pipeline, but it is not validated. The KG is built with LightRAG — generic LLM-based entity extraction with no domain ontology — yet no extraction precision/recall is reported. There is also no ablation comparing KG filtering against simpler alternatives (e.g., direct LLM fact-checking) or no filtering. The added value of the KG is unclear.
3. All experiments use Qwen2.5-VL-7B and XLRS-Bench only. It is unclear whether the findings generalize to other MLLMs (e.g., Qwen3-VL, LLaVA, or InternVL) or other UHR RS benchmarks (e.g., RSHR [5]).
4. Table 4 compares the proposed method (zoom-in tool + 32-sample budget) against closed-source models evaluated with default API settings (likely single-shot, no tool). This makes the SOTA claim potentially misleading.

### References

- [1] Zheng et al., "DeepEyes: Incentivizing Thinking with Images via Reinforcement Learning", arXiv:2505.14362, 2025.
- [2] Chen et al., "The Synergy Dilemma of Long-CoT SFT and RL: Investigating Post-Training Techniques for Reasoning VLMs", arXiv:2507.07562, 2025.
- [3] Chen et al., "Advancing Multimodal Reasoning: From Optimized Cold Start to Staged Reinforcement Learning", arXiv:2506.04207, 2025.
- [4] Peng et al., "LMM-R1: Empowering 3B LMMs with Strong Reasoning Through Two-Stage Rule-Based RL", arXiv:2503.07536, 2025.
- [5] Dang et al., "A Benchmark for Ultra-High-Resolution Remote Sensing MLLMs", arXiv:2512.17319, 2025.

---

> ### Author Rebuttal · Authors · 2026-03-30
>
> ## W1:Clarify difference with [1, 2, 3, 4].
> Thanks for the thoughtful comment. In fact, reference [1] serves as our baseline, while [2] and [3] are already cited in Lines 475 and 482 of the main paper.
>
> We would like to address your concern at a high level. (1) It is well understood that new data and methods such as Agentic RLVR can improve performance. (2) However, in a scientifically grounded domain like remote sensing, simply transplanting standard-domain methods or naively adding new data does not produce effective gains. (3) We further studied how to improve the method and introduce new data effectively, and found that what actually makes remote sensing MLLMs work differs substantially from common intuition. Beyond point (1), these methodological and counterintuitive findings in points (2) and (3) form the core contribution of our paper.
>
> Below, we will respond to your each point in turn.
>
> **1. Conclusion 1 originates from DeepEyes?**
>
> With multiple post-training methods now emerging, a key question for the remote sensing community is which method is actually best suited to UHR remote sensing training. This cannot be directly answered by citing DeepEyes alone; it requires controlled ablations in the remote sensing domain. More importantly, we do not simply restate the advantage of Agentic RLVR. Our ablations reveal both its strengths and its limitations in the remote sensing setting.
>
> * **Pass@1.** As shown in Figure 3(a), directly training Agentic RLVR with remote sensing data causes a **2.31%** drop in Pass@1. This suggests that Agentic RLVR is not sufficiently stable on UHR remote sensing data and cannot effectively learn remote sensing VQA during the RL stage. This domain-specific finding directly motivates Conclusions 2 and 3: how to enable Agentic RLVR to better absorb remote sensing domain knowledge. This phenomenon is not reported in DeepEyes.
>
> * **Pass@32.** DeepEyes does not examine Pass@32, which reflects the upper bound of reasoning ability. In contrast, we explicitly analyze this metric and show the advantage of Agentic RLVR at the reasoning boundary in the UHR remote sensing setting.
>
> Overall, the performance we observe in both Pass@1 and Pass@32 differ from those reported in DeepEyes.
>
> **2. Conclusion 3 is already well known [2, 3, 4]?**
>
> Although our conclusion may appear similar to [2, 3, 4], it is fundamentally different. Our contribution is not merely that domain text is more useful than generic text for remote sensing. More importantly, we show that how to construct high-quality domain text, and we provide a feasible solution through our knowledge-graph(KG) guided pipeline.
>
> This distinction is crucial:
> * **Generic reasoning text does not help UHR remote sensing.** Reference [2] shows that using s1.1-R1 dataset for SFT improves multimodal reasoning. However, in our experiments, replacing our Earth-science text with the s1.1-R1 leads to clear performance degradation. This suggests that the key is not whether text is added, but whether it is carefully constructed to match the target domain’s knowledge structure and reasoning needs.
>
> |Text SFT|SuperRS|RL(Gen+RS)|Pass@1|Pass@32|
> |-|-|-|-|-|
> |ES|✓|✓|60.40|96.25|
> |s1.1-R1|✓|✓|46.98|90.87|
>
> * **Building high-quality domain text with our KG-based pipeline**. We will add a ablation study, while varying only the quality-control method of the Earth-science text QA:
> (a) minimal-cleaning text
> (b) full-filtered text with our KG-based validation
>
> |ES text|Filter|Pass@1|Pass@32|
> |-|-|-|-|
> |30k|KG|55.4|92.9|
> |30k|None|52.2|91.3|
> |148k|KG|60.4|96.3|
> |148k|None|58.9|94.8|
>
> We would be glad to continue the discussion.
>
> ## W2:Ablation of KG-based filtering
> Thanks for your comments. Considering the limited responce space, please refer the Responce the **Reviewer-urbT Weakness 4.** We found that adding domain text itself already provides substantial benefit, while stronger filtering further improves the magnitude and stability of that benefit.
>
> ## W3:Other base models or other UHR RS benchmarks?
> 1. **Other base models.** Due to time constraints, we were only able to repeat the experiment on Qwen3-VL.
>
> |Method|Pass@1|Pass@32|
> |-|-|-|
> |Qwen3-VL-8B|50.02|88.73|
> |Baseline|56.73|92.28|
> |+pre-warm|58.76|94.19|
> |+pre-warm+ES text|63.15|97.32|
>
> 2. **Other benchmarks.** Since RSHR, released on December 19, 2025, has not yet undergone rigorous peer review, we chose LRSVQA (ICCV 2025) to evaluate our strategy. We believe RSHR is also an excellent work and we will cite and discuss it in more detail in the related work section. We have added the new results in the responce to the **Reviewer-ztUV Weakness2-3.** Our method still perform better than other works in LRSVQA.
>
> ## W4:Evaluate closed-source models.
> Thank you again for your comment. Due to space limitations, we cannot elaborate further here, so we refer you to the end of our response to **Reviewer-9bTn**, where we provide a more detailed discussion.

---

> > ### Author Rebuttal · Reviewer_gvqX · 2026-04-04
> >
> > I thank the authors for their effort in the rebuttal. I acknowledge this work as one of the first in-depth investigations of SFT + RLVR in the remote sensing domain and therefore increase my score.

---

> > > ### Author Response · Authors · 2026-04-04
> > >
> > > Your recognition of our work as a pioneering effort in the remote sensing domain is deeply encouraging to us.We will incorporate the rebuttal results and discussions into the final manuscript. Thank you again for your insightful feedback and support.

---

### Official Review · Reviewer_urbT · 2026-03-10

**Soundness:** 3
**Presentation:** 4
**Significance:** 2
**Originality:** 3
**Overall Recommendation:** 5
**Confidence:** 4

**Summary:**

This paper compares post-training techniques: supervised fine-tuning, RLVR, and Agentic RLVR with a zoom-in tool, for understanding ultra-high-resolution remote sensing images. Surprisingly, training on Earth-science text alone (no images) proves to be the biggest performance driver. Based on this, the authors propose a staged training recipe: start with text-only QA to build domain knowledge, then warm up on hard image examples before applying reinforcement learning. This allows a small 7B model to outperform much larger models on the XLRS-Bench benchmark.

**Compliance With Llm Reviewing Policy:**

Affirmed.

**Final Justification:**

Thank you for the detailed follow-up. The corpus description, covering 8,848 textbooks across all Earth-system spheres with a clear three-stage processing pipeline, directly addresses my remaining concern about transparency. This is the information needed to evaluate and reproduce the central finding. Generalization beyond remote sensing remains an open question, but as noted earlier, this is an acknowledged limitation rather than a flaw in the work. I am satisfied with the authors' responses and maintain my score of 5.

**Key Questions For Authors:**

The related work section is placed just before the conclusion, which disrupts the flow of the paper. Conventionally, related work follows the introduction to help readers contextualize the contribution before diving into the technical details. The authors should consider moving it earlier.

**Limitations:**

1. The proposed recipe is only tested on remote sensing data, leaving its generalizability to other domains an open question. Testing on other high-resolution image domains, such as histopathology, would strengthen the claim that this is a broadly applicable training strategy.
2. The results section would benefit from a qualitative example showing how the model reasons through a real satellite image, which would make the improvements more concrete and easier to interpret.
3. The three-stage pipeline is considerably more expensive than standard fine-tuning, yet the paper reports no training time, GPU hours, or data requirements for any stage. This makes it hard for practitioners to assess whether the gains are worth the added cost, or whether simply scaling up a larger model would be a more practical alternative.

**Strengths And Weaknesses:**

Strengths:
1. Despite being a simple idea, the paper is well-written and easy to follow. The figures are clear and effectively support the narrative.
2. The staged knowledge injection recipe is a practical and useful contribution that others can directly build on.
3. The paper is well-written and easy to follow despite covering a fairly complex training pipeline.

Weaknesses:
1. Evaluation is limited to a single benchmark, raising questions about reliability. XLRS-Bench has a specific distribution of tasks, image types, and question formats, and it's possible the staged recipe is inadvertently optimized for its quirks rather than for UHR reasoning in general.
2.  Generalization beyond remote sensing is assumed but not demonstrated.
3. No qualitative analysis of failure cases or model reasoning
4. The paper's central claim hinges on Earth-science text-only QA being the key performance driver, yet the dataset itself is barely described. This is a significant gap, because the surprising result might simply reflect that this text dataset is unusually high-quality rather than anything special about text-only training. Without these details, the finding is difficult to interpret or reproduce.

---

> ### Author Rebuttal · Authors · 2026-03-30
>
> We sincerely thank your highly positive assessment of our paper and for your constructive suggestion. We will address your insightful comments point by point in the following discussion.
> ## Limitation 1: Other high-resolution image domains.
> Thank you very much for this valuable suggestion. We agree that extending our framework to other domains is both meaningful and feasible. However, fully carrying out such an extension during the rebuttal period is not practical, as it would require rebuilding the entire pipeline for a new domain.
>
> More specifically, our method is not limited to remote sensing at the methodological level. Core components of our framework, including domain-specific knowledge corpus construction, data curation and cleaning, and task-oriented training data generation,can in principle be applied to other scientific domains, such as histopathology.
>
> Although we do not have sufficient time in the rebuttal phase to fully extend the complete workflow to a new domain, we have added new results on the LRSVQA (ICCV 2025) benchmark, which further demonstrate the effectiveness of our method on an additional remote sensing benchmark. These results are reported in our response to **Reviewer-ztUV Weakness2-3.**
>
> ## Limitation 2: A qualitative example.
> Thank you very much for your constructive suggestion. Unfortunately, OpenReview does not allow us to include images. However, your suggestion is highly valuable, and we will incorporate a qualitative example in the camera-ready version to illustrate our reasoning performance.
>
> ## Limitation 3: Report  training time, GPU hours, or data requirements.
> Thanks your insightful comments. Our training pipeline consists of only two stages: standard SFT and Agentic RLVR. Below, we provide the data used in each stage and the corresponding training time required.
>
> |Stage|Data|HW|Time|
> |-|-|-|-|
> |SFT|148k earth-science text<br/>+12k RS-VQA|2×8 A100|~2h|
> |Agentic RLVR|47k general VQA<br/>+12k RS-VQA|2×8 A100|~14h|
>
> The table below compares our method with other strong baselines.
>
> |Method|Train|HW|Time|XLRS-Bench|
> |-|-|-|-|-|
> |GeoLLaVA-8K|SFT|2×8 A100|~10h|51.50|
> |DeepEyes|Agentic RLVR|2×8 A100|~27h|52.39|
> |Ours|SFT+Agentic RLVR|2×8 A100|~16h(2+14)|60.40|
>
> Compared with standard SFT+RL pipelines, Text-Before-Vision adds little computational overhead.
> * Compared with GeoLLaVA-8K, which uses 81k UHR VQA samples during SFT, our method uses only 12k VQA samples plus pure text data, making the SFT stage much shorter. Although adding Agentic RLVR increases our total training time by about 6 hours, the resulting performance gain is substantial.
> *  Compared with DeepEyes, which trains for more than 64 RL steps, our method uses only 24–32 RL steps and adds a lightweight SFT stage of about 3 hours, keeping the overall training time well below that of DeepEyes.
>
> Overall, our method achieves substantially better performance with less training time than the baseline methods.
>
> Your insightful comments also concern whether simply scaling up model size might be more practical. We believe improving data quality and training strategy is far more effective than increasing parameter count alone. As shown in Table 4 in our main text, the 8B Intern-S1-Mini achieves 51.6%, while scaling to **Intern-S1 (241B)** raises Pass@1 to only **55.0%**. In contrast, our improvements in data and training strategy enable a 7B model to reach **60.40%**. Our approach is therefore clearly more economical.
>
> ## Weakness 4: the dataset itself is barely described
> Thank you for your comment.
> 1. Regarding your comments that "results may simply reflect the unusually high quality of the text dataset", our claim is that staged domain knowledge injection in text form is itself a major driver of improvement, rather than performance gains arise merely because we happened to curate an unusually strong text corpus. To address this concern, we will add a controlled ablation, while varying only the quality-control strength of the Earth-science text QA:
> (a) minimal-cleaning text
> (b) full-filtered text with our knowledge-graph-based validation
>
> |ES text|Filter|Pass@1|Pass@32|
> |-|-|-|-|
> |30k|KG|55.4|92.9|
> |30k|None|52.2|91.3|
> |148k|KG|60.4|96.3|
> |148k|None|58.9|94.8|
>
> We found that adding domain text itself already provides substantial benefit, while stronger filtering further improves the magnitude and stability of that benefit.
>
> 2. What's more, we have already provided an overview of the dataset characteristics in Section 4 of the main paper, and Table 2 presents the main statistics of our dataset. In addition, Appendix B, *Data Pipeline* and Table 5, *Task Hierarchy and Definitions*, presents our task categories together with representative examples.
>
> We will report these results and expand the dataset description in the revised paper.
>
> **We sincerely thank your highly positive comments again and please do not hesitate to contact us with any further questions.**

---

> > ### Author Rebuttal · Reviewer_urbT · 2026-04-02
> >
> > Thanks for the detailed rebuttal. The training cost question is fully addressed; the comparison table is clear and convincing.
> > On the dataset concern, the new ablation is helpful but doesn't quite close the gap. It shows that scale and filtering matter, but I still don't know what's actually in the Earth-science corpus, sources, topic coverage, etc. That's the piece needed to evaluate whether the central finding generalizes beyond this specific dataset.
> > The generalization concern remains open. LRSVQA is still a remote sensing benchmark, so adding it doesn't address whether the staged recipe works outside this domain. I understand a full extension isn't feasible during rebuttal; I just want this noted as a real limitation of the current submission.
> > Maintaining my score of 5. The contribution is solid, but I'd encourage the authors to be more transparent about the corpus in the final version.

---

> > > ### Author Response · Authors · 2026-04-03
> > >
> > > Thank you again for your constructive comments and positive feedback.
> > >
> > > **1. Earth-science corpus, sources, topic coverage.**
> > > We collected **8,848 Earth-science textbooks and monographs** in PDF format, covering authoritative materials across all Earth-system spheres.
> > >
> > > |Sphere|Field|Count|Storage/MB|
> > > |-|-|-|-|
> > > |Atmosphere|Atmospheric science|544|19152|
> > > |Atmosphere|Atmospheric chemistry|66|1429|
> > > |Atmosphere|Geography|750|36605|
> > > |Atmosphere|Meteorology|105|3483|
> > > |Atmosphere|Hydrology|404|20746|
> > > |Atmosphere|Hydrometeorology|13|167|
> > > |Atmosphere|Paleoclimatology|20|668|
> > > |Biosphere|Biogeochemistry|93|1896|
> > > |Biosphere|Biogeography|206|5698|
> > > |Biosphere|Ecology|750|29279|
> > > |Biosphere|Landscape ecology|117|3876|
> > > |Biosphere|Archaeology|38|1305|
> > > |Biosphere|Palynology|5|73|
> > > |Biosphere|Paleomicrobiology|12|513|
> > > |Hydrosphere|Hydrology|323|8818|
> > > |Hydrosphere|Hydrogeology|139|5118|
> > > |Hydrosphere|Limnology|63|1692|
> > > |Hydrosphere|Oceanography|269|8553|
> > > |Hydrosphere|Marine chemistry|12|341|
> > > |Hydrosphere|Physical oceanography|50|1352|
> > > |Hydrosphere|Marine biology|7|162|
> > > |Hydrosphere|Marine geology|9|205|
> > > |Lithosphere|Geology|56|1661|
> > > |Geosphere|Economic geology|25|1091|
> > > |Geosphere|Engineering geology|81|6409|
> > > |Geosphere|Environmental geology|36|4043|
> > > |Geosphere|Forensic geology|1|21|
> > > |Geosphere|Historical geography|21|1581|
> > > |Geosphere|Quaternary geology|13|426|
> > > |Geosphere|Planetary geology|5|118|
> > > |Geosphere|Sedimentology|95|5115|
> > > |Geosphere|Stratigraphy|218|9347|
> > > |Geosphere|Human geography|247|14902|
> > > |Geosphere|Physical geography|130|10718|
> > > |Geosphere|Geochemistry|358|13699|
> > > |Geosphere|Geomorphology|229|12263|
> > > |Geosphere|Geophysics|375|20278|
> > > |Geosphere|Geochronology|21|316|
> > > |Geosphere|Geodynamics|138|5194|
> > > |Geosphere|Structural geology|247|13290|
> > > |Geosphere|Geomagnetism|19|998|
> > > |Geosphere|Gravimetry|13|212|
> > > |Geosphere|Geodesy|159|3730|
> > > |Geosphere|Seismology|135|6523|
> > > |Geosphere|Glaciology|11|1026|
> > > |Geosphere|Hydrogeology|139|5118|
> > > |Geosphere|Mineralogy|187|10334|
> > > |Geosphere|Crystallography|331|6106|
> > > |Geosphere|Gemology|14|1241|
> > > |Geosphere|Petrology|94|5440|
> > > |Geosphere|Rock physics|25|993|
> > > |Geosphere|Speleology|8|343|
> > > |Geosphere|Volcanology|20|1336|
> > > |Pedosphere|Soil science|152|6349|
> > > |Earth system|Remote sensing|750|31512|
> > > |Earth system|Geostatistics|107|2298|
> > > |Earth system|GIScience|42|872|
> > > |Earth system|Cartography|304|15692|
> > > |Earth system|Geoinformatics|47|1261|
> > >
> > > We process the collected textbooks in three stages: (1) OCR extraction via MinerU; (2) LLM-based preliminary cleaning; (3) LLM-based enhancement.
> > >
> > > (1) OCR extraction using MinerU. MinerU extracts the main body text reliably, and we do not elaborate on the framework here given its broad usage and maturity. Completing the OCR extraction alone took nearly 7 days on 32× A100 GPUs.
> > >
> > > (2) LLM-based preliminary processing. Because the raw scanned documents contain a variety of structural and formatting issues, we apply:
> > > * removal of Emails, links;
> > > * Unicode repair;
> > > * normalization of Chinese and English punctuation;
> > > * SimHash-based deduplication;
> > > * segmentation of the text into chunks no longer than 4096 tokens (to facilitate downstream LLM operations).
> > >
> > > (3) LLM-based enhancement. Even after step (2), the scanned documents still contain common textual issues—missing characters, broken sentences, inconsistent phrasing, and local incoherence. We therefore perform an additional round of LLM-based linguistic correction, focusing strictly on improving readability and continuity without introducing any new information. Even with 32× A100 GPUs, the processing required nearly 10 days to complete.
> > >
> > > Based on the resulting high-quality data, as shown in Figure 5 of the main paper and Table 5 of the appendix, we then use an LLM to generate diverse question-answer pairs.
> > >
> > > **2.Generalization**
> > > We strongly agree that generalization is an important concern in the future. We are also interested in applying similar data construction pipelines to domains such as medicine, chemistry, and biology to test whether domain-specific text can benefit perception and reasoning. Motivated by your comment, we have already begun reviewing related efforts in other fields.
> > >
> > > For example, *A Family of Large Language Models for Materials Research with Insights into Model Adaptability in Continued Pretraining* (**Nature Machine Intelligence, Feb. 27, 2026**) uses about 4 million materials-science publications and crystallographic data. However, that work mainly uses domain text for pretraining, and does not study (1) how such text affects the reasoning boundary, or (2) when and how this knowledge should be injected most effectively. Our work explores these questions in Earth science. Building on this materials work, we plan to test whether our conclusions extend to materials science and, even more broadly, to other disciplines, with the longer-term goal of studying possible synergies across domain-specific text corpora.
> > >
> > > We are truly grateful for your recognition and deeply encouraged by comments.

---

### Official Review · Reviewer_ztUV · 2026-03-13

**Soundness:** 2
**Presentation:** 3
**Significance:** 2
**Originality:** 3
**Overall Recommendation:** 4
**Confidence:** 3

**Summary:**

This paper introduces a novel "Text-Before-Vision" training paradigm to tackle the bottleneck of visual evidence acquisition in ultra-high-resolution (UHR) remote sensing (RS). Because standard reinforcement learning often struggles to navigate massive UHR pixel spaces without structured domain priors, the authors propose a staged knowledge injection approach that uniquely leverages text-only data to drive multimodal reasoning. They build a automated data pipeline for generation of large-scale (148K pairs) Earth-science text-only QA dataset. Evalution on XLRS-Bench demonstrate the robustness of their approach.

**Compliance With Llm Reviewing Policy:**

Affirmed.

**Final Justification:**

The rebuttal has resolved my concerns and I have adjusted the scores accordingly.

**Key Questions For Authors:**

1. What is the computational overhead and processing time associated with the Knowledge Graph cross-validation step when generating the 148,777 QA pairs?
2. How sensitive is the Agentic RLVR stage to the specific hyperparameters of the reward design (e.g., the conditional tool bonus) when transitioning from the cold-start SFT model?

**Limitations:**

Yes.

**Strengths And Weaknesses:**

Strengths
1. The paper is novel in using counter-intuitive text-only Earth-science QA data.
2. The paper introduce a data comprehensive and automated data generation pipeline that extracts information from both broad textbooks and scientific papers which leads to large-scale (148K pairs) Earth-science text-only QA dataset.
3. Empirical results on XLRS-Bench demonstrate its performance.

Weakness
1. The paper's focus is on a niche area, with new data generated in this domain would not surprising bring improvement to the model.
2. The evaluation is also very limited.
3. The model could be overfitted on this single task.

---

> ### Author Rebuttal · Authors · 2026-03-30
>
> ## Weakness1: Niche area, with new data generated in this domain would not surprising bring improvement.
>
> Thanks for your comments. Ultra-high-resolution remote sensing (RS) has attracted increasing attention in recent years. For instance, the Nature paper "Observing the tidal pulse of rivers from wide-swath satellite altimetry" (March 18, 2026) underscores the scientific value of wide-swath satellite imagery. Meanwhile, RS has become increasingly prominent at conferences. NeurlPS 2026 has accepted Pan-LUT（focus large remote sensing images）as oral, CVPR includes a dedicated remote sensing field. These trends indicate that the RS field has both a strong community foundation and clear research significance.
>
> What's more, our contribution is not as simply introducing new data into a relatively small field and therefore obtaining performance gains.
>
> * Simply adding more data does not necessarily produce positive effects in this field. This is widely recognized both in RS and in general LLMs community: data quality, rather than sheer scale, is often the key factor driving progress. Therefore, we believe that creating new, high-quality data is a meaningful community contribution.
>
> * More importantly, however, our contribution is not merely to produce data, but to identify what kinds of remote sensing data actually improve performance. Rather, our main contributions are twofold: (1) we show that simply transferring standard-domain methods or adding new data does not yield meaningful gains; and (2) we study how to adapt methods and introduce data effectively for a scientific domain such as remote sensing, leading to new findings and insights.
>
> Overall, our goal is not simply to generate new data, but to understand, within a scientifically important field, which post-training method and which forms of data can most effectively move the community forward.
>
>
> ## Weakness2-3: Limited evaluation, overfitted on this single task.
> We have evaluated our strategy on 13 sub-tasks in XLRS-Bench, including counting, localization, object relations, environmental reasoning and so on, and observed consistent gains across diverse task types.
>
> To further broaden the evaluation, we also tested our strategy on LRSVQA (ICCV 2025), a large-format remote sensing benchmark with eight sub-tasks.
>
> |Method|Size|FAIR|Bridge|STAR|Avg|
> |-|-|-|-|-|-|
> |Qwen3-VL|8B|27.98|38.56|32.04|32.86|
> |InternVL3.5|8B|25.14|35.50|26.86|29.17|
> |RSCoVLM|7B|27.37|42.42|31.77|33.85|
> |RSCoVLM+ZC|7B|42.42|49.56|45.15|45.71|
> |Ours|7B|44.50|51.22|47.79|47.84|
>
> ## Q1：Computational overhead of the Knowledge Graph cross-validation step.
> Thank you for your comment. Although constructing the knowledge graph (KG) required substantial computational resources, using the Earth science KG for cross-validation is comparatively inexpensive. We also plan to open-source the KG, so the community will not need to rebuild it.
>
> 1. **Knowledge graph construction.** We used LightRAG to build a structured graph from our cleaned textbook corpus, yielding 62,996 nodes and 79,967 edges. This stage required about 40 hours on a node with 8 NVIDIA A100 GPUs.
>
> 2. **Cross-validation.** KG-based cross-validation for candidate QA pairs is cheaper. On 8×A100 (80GB) GPUs, this process took about 10 hours and filtered out roughly half of the candidate QA pairs, leaving 148,777 high-quality samples. For each candidate QA pair, we used the question (Q) as a query and performed two-level retrieval with LightRAG over the KG, at both the entity and text-chunk levels, to obtain supporting evidence and similarity scores. We applied a similarity threshold to filter out out-of-domain or unsupported hallucinated QA pairs and finally retained 148,777 QA pairs.
>
> ## Q2：Hyperparameters of the reward design.
> Thank you for your comment. To clarify, our paper does not claim any contribution to reward design in the RLVR stage. Rather, our contribution lies in a domain knowledge injection strategy. Accordingly, we kept the default hyperparameters of the DeepEyes baseline unchanged and made no task-specific modifications to the RL reward design. We appreciate your suggestion and conducted an ablation study on the tool_bonus weight to examine whether it affects our main conclusions.
>
> |tool_bonus|pass@1|pass@32|
> |-|-|-|
> |1.2(default)|60.40|96.25|
> |1.5|**60.54**|**96.57**|
> |0.6|59.54|95.89|
>
> Across different settings, Pass@1 and Pass@32 varied by less than 1%, indicating that the overall results are highly stable. The original DeepEyes paper likewise reports a slight performance drop when this term is removed. Overall, neither the use of this hyperparameter nor changes to its weight affect our core conclusion about the value of domain knowledge data.

---

> > ### Author Rebuttal · Reviewer_ztUV · 2026-03-31
> >
> > My concerns are resolved and I have adjusted the scores accordingly.

---

> > > ### Author Response · Authors · 2026-04-04
> > >
> > > Thanks for your positive response and for letting us know that your concerns are fully resolved. We deeply appreciate the time and effort you dedicated to reviewing our work.

---

### Decision · Program_Chairs · 2026-04-30

**Decision:**

Accept (regular)

**Comment:**

This paper was reviewed by four experts in the field. The recommendations are (Weak Accept x 2, Accept, Strong Accept). Based on the reviewers' feedback, the decision is to recommend the acceptance of the paper. The reviewers did raise some valuable concerns (especially more detailed experimental comparisons and ablation studies raised by all four Reviewers, clearer paper presentation and statement raised by Reviewer gvqX and 9bTn) that should be addressed in the final camera-ready version of the paper. The authors are encouraged to make the necessary changes to the best of their ability.